# Current Landscape of Immune Checkpoint Inhibitor Therapy for Hepatocellular Carcinoma

**DOI:** 10.3390/cancers14082018

**Published:** 2022-04-16

**Authors:** Nikolaos Machairas, Diamantis I. Tsilimigras, Timothy M. Pawlik

**Affiliations:** 12nd Department of Propaedeutic Surgery, National and Kapodistrian University of Athens, 11527 Athens, Greece; nmachair@med.uoa.gr; 2Department of Surgery, Division of Surgical Oncology, The Ohio State University Wexner Medical Center and James Comprehensive Cancer Center, Columbus, OH 43210, USA; diamantis.tsilimigras@osumc.edu; 3Department of Surgery, The Urban Meyer III and Shelley Meyer Chair for Cancer Research, The Ohio State University Wexner Medical Center, Columbus, OH 43210, USA

**Keywords:** hepatocellular carcinoma, liver cirrhosis, immunotherapy, immune checkpoint inhibitors

## Abstract

**Simple Summary:**

Hepatocellular carcinoma (HCC) is the most common primary hepatic malignancy, with increasing incidence over the past several decades. The majority of patients with HCC present with advanced unresectable disease, making treatment options with curative intent limited and survival outlooks dismal. Systemic therapy with sorafenib had been traditionally used, with marginal benefit. Immunotherapy, successfully used to treat other malignant tumors, has recently been shown to be safe and well tolerated and to have promising long-term outcomes in patients with advanced HCC. We herein review the outcomes of immune checkpoint inhibitors (ICI) from major clinical trials, summarize predictors of treatment response, and highlight adverse events related to ICI treatment.

**Abstract:**

Hepatocellular carcinoma (HCC) is the most frequent primary liver tumor. As a result of advanced disease being often present at diagnosis, only a small percentage of patients are amenable to curative-intent treatment options such as surgical resection and liver transplantation. Systemic therapy consisting of tyrosine kinase inhibitors such as sorafenib had been used for over a decade with limited efficacy. More recently, treatment with immune checkpoint inhibitors has revolutionized the treatment landscape of various malignant tumors. With this shifting paradigm, recent data have demonstrated encouraging outcomes among patients with HCC. In particular, several trials have investigated the safety and efficacy of various immune checkpoint inhibitors (ICI) either as monotherapy or in the form of combined treatments. We sought to provide an overview of recent clinical trials among patients with advanced HCC as well as to highlight predictors of response and immune-related adverse events and to review the evidence on perioperative administration of ICI in patients with resectable HCC.

## 1. Introduction

Hepatocellular carcinoma (HCC) remains the most commonly encountered primary hepatic tumor [1]. At present, HCC is the sixth most frequent cancer type and the third leading cause of cancer-related death worldwide [1]. The reported incidence of HCC ranges substantially worldwide, from 2.4 to 13.4 per 100,000, as a consequence of geographic variability in the prevalence of risk factors [2,3]. HCC develops principally in the context of cirrhosis. The main causative risk factor for development of liver cirrhosis, and accordingly HCC, is chronic hepatitis B virus infection (HBV), while other significant risk factors include hepatitis C virus infection (HCV), alcoholic liver disease, non-alcoholic fatty liver disease (NAFLD), and non-alcoholic steatohepatitis (NASH) [3,4,5].

The optimal management of patients with HCC is tailored in the context of a multidisciplinary team, and is based on disease and patient characteristics including the presence of cirrhosis, disease stage, and underlying liver function and performance status. The Barcelona Clinic Liver Cancer (BCLC) staging classification system first proposed over two decades ago, as well as its more recent revisions, has been widely used for stratification and treatment allocation of cirrhotic patients with HCC [6,7]. Despite the fact that surgical treatment options, including liver resection and transplantation, remain the mainstay for selected patients (BCLC 0 and A, respectively) yielding the best chance at overall survival, only a relatively small subset of patients are eligible for these options [8,9,10]. This is mainly due to advanced disease or poor performance status at presentation, and HCC is accordingly associated with dismal survival. Patients with more advanced disease (BCLC B) are treated with locoregional treatments, including transarterial chemoembolization, while more than 80% of HCC patients present at an even more advanced stage (BCLC stage C). For these patients, curative and locoregional treatment options are not suitable, and thus survival is poor, reportedly as low as 15%. 

Sorafenib was the first targeted systemic therapy regimen shown to be effective in patients with advanced HCC (aHCC) and was used as standard therapy for over a decade. The SHARP trial first demonstrated that systemic treatment with sorafenib treatment extended the median overall survival (OS) of patients compared with placebo (10.7 vs. 7.9 months; hazard ratio (HR) 0.69; 95% Confidence intervals (CI) 0.55–0.87; *p* < 0.001) [11]; similar results were demonstrated in the Phase 3 Asia-Pacific study [12]. Over the past several years, the treatment landscape for patients with aHCC has broadened with the approvals of novel orally-administered tyrosine kinase inhibitors (TKIs) including lenvatinib, regorafenib, and cabozantinib, as well as immunotherapies such as immune checkpoint inhibitors (ICI) including atezolizumab, nivolumab, and pembrolizumab. According to the most recent 2022 version of the BCLC, the combination of atezolizumab with bevacizumab (Atezo-Bev) is currently the first-line treatment for patients with aHCC, as it confers a superior survival benefit versus sorafenib [7].

## 2. Immune Checkpoint Inhibitors

During the past decade, numerous trials have demonstrated the beneficial role of immunotherapy in terms of improved OS for various malignant tumors, including non-small cell lung, pancreatic, gastric and esophageal cancer [13,14,15,16]. Oncolytic immunotherapy has emerged as a promising approach to inhibiting tumor progression and metastasis. This approach is based on the enhancement of cellular or humoral immunity through activation of tumor-specific immune responses, breaching immune tolerance [17]. Of note, immunotherapy has been demonstrated to be safe and largely well-tolerated. Immunotherapy reverses the tumor-expressed extracellular ligands, which subdue intrinsic immune response, with cytotoxic T-lymphocyte-associated antigen 4 (CTLA-4, sometimes known as CD152) and programmed cell death protein-1 (PD-1 or CD279) as well as ligand PD-L1 being the classical examples.

The liver is considered to have a highly complicated immune tolerance system driven by antigen-presenting cells (APC), which actively modulate immunogenicity of the liver microenvironment [17,18,19]. APCs generate signals for immune checkpoint molecules in order to inhibit hyperactivation of T cells [18]. These suppressive signals from immune checkpoint molecules play a pivotal role in maintaining tolerance and preventing unwanted immune responses, which can lead to tissue damage. Malignant tumor microenvironments (TME) perturb normal suppressive signals, causing T-cell exhaustion. In turn, this can cause hyperexpression of immune checkpoint molecules, impaired cytotoxicity, and diminished levels of effector cytokines [17,20]. This process eventually leads to chronic hyporesponsive immunity [20]. Taking into account the significant regulatory role of immune checkpoint molecules in immune tolerance, a large number of clinical trials have aimed to confirm their function and efficacy in the treatment of HCC.

ICI are monoclonal antibodies (mAb), which block extracellular proteins, which in turn inhibits the antitumor immune response. ICIs are increasingly widely used and have had demonstrated beneficial effects in treating a number of advanced malignant tumors, including aHCC (Table 1). To date, two specific immune-mediating molecules have been extensively used in clinical trials, namely, PD-L1 and CTLA-4. These agents, the beneficial action of which has been demonstrated by high quality trials, have been included in the most recent treatment algorithm of the BCLC for patients with advanced-stage HCC (BCLC B-C) [7]. 

Currently, based on the European Association for the Study of the Liver (EASLD) recommendations, atezolizumab/bevacizumab is the preferred treatment option for patients with aHCC eligible for systemic therapy [21]. Moreover, remucinumab is recommended as a second-line treatment option for patients who have been previously treated with sorafenib, along with alpha-fetoprotein (AFP) ≥ 400 ng/dL. Additionally, nivolumab or pembrolizumab monotherapy as well as combined treatment with nivolumab and ipilimumab should be considered as second-line treatment based on data from several single-arm trials.

### 2.1. Atezolizumab

Atezolizumab is a fully humanized IgG1 mAb engineered with a modification in the Fc domain. This modification facilitates elimination of antibody-dependent cellular cytotoxicity, thus preventing depletion of T cells which express PD-L1 [22]. Atezolizumab has been demonstrated to increase the level of proliferating CD8^+^ T cells. This effect is achieved by inducing a plethora of cytokine changes, including transient increases of IL-18, IFNγ, and CXCL11 and short-term inhibition of IL-6 [23]. Atezolizumab reduces immunosuppressive signals located within the tumor microenvironment by inhibiting PD-L1 and subsequently increasing T cell-mediated immunity against several tumors, including HCC. 

An open-label, multicentre, multi-arm, phase 1b study included 119 patients with unresectable HCC randomized to treatment with atezolizumab plus bevacizumab or monotherapy with atezolizumab [24]. Patients receiving combination therapy had improved progression-free survival versus monotherapy (median 5.6 vs. 3.4 months, HR 0.55; 80% CI 0.40–0.74; *p* = 0·011), while median OS was not reached in either treatment group [24]. 

The much awaited open-label phase III IMbrave150 (NCT03434379) trial evaluated the efficacy of atezolizumab plus bevacizumab in combination compared with sorafenib therapy as a first line treatment in 501 patients with aHCC [25]. Patients treated with atezolizumab plus bevacizumab had significantly improved twelve-month OS versus patients treated with sorafenib (67.2% vs. 54.6%, respectively). Additionally, the median progression-free survival (PFS) was 6.8 months in the combination group versus 4.3 months in the sorafenib group (HR for disease progression or death 0.59; 95% CI, 0.47 to 0.76; *p* < 0.001) [25].

### 2.2. Nivolumab

Nivolumab is another fully human IgG4 PD-1 mAb that blocks PD-1 and facilitates restoration of anticancer immune responses by abrogating PD-1 pathway-mediated T-cell inhibition. Administration of nivolumab was first authorized in 2017 for treatment of patients with aHCC as an effective alternative for patients with progressive disease after first-line treatment with sorafenib. In the first major non-comparative open label phase I/II international trial (CheckMate 040) [26], researchers evaluated the impact of nivolumab administration in patients with aHCC irrespectively of HBV/HCV infection and whether treated previously with sorafenib or not. Patients were required to have a Child Pugh (CP) score of ≤7 (CP A or B7) for the dose-escalation phase and ≤6 or less (CP A) for the dose-expansion phase, as well as an Eastern Cooperative Oncology Group performance status (ECOG PS) ≤1. A total of 262 patients were included. Grade 3/4 adverse events (AE) related to nivolumab administration were observed in 25% of patients, while only 6% had serious AE. A considerable objective response rate (ORR) of 15% was demonstrated. Moreover, the disease control rate and the median time to progression were 58.5% and 3.4 months, respectively. The observed median OS for patients in the dose-escalation phase was fifteen months (95% CI 9.6–20.2) [26]. 

More recently, CheckMate 459, an international phase III randomized controlled trial (RCT), aimed to evaluate nivolumab monotherapy compared with sorafenib monotherapy in a first-line setting for patients with aHCC [27]. The trial included patients with histologically confirmed aHCC not eligible for or with progressive disease following surgical resection or use of locoregional treatment and with no previous systemic therapy for HCC; patients needed to be Child-Pugh class A and have an ECOG PS score of 0 or 1 regardless of viral hepatitis status. A total of 743 patients were randomly assigned to treatment with nivolumab (*n* = 371) or sorafenib (*n* = 372). The median OS for patients who received nivolumab and sorafenib was 16.4 (95% CI 13.9–18.4) and 14.7 months (11.9–17.2), respectively (hazard ratio 0.85, 95% CI 0.72–1.02, *p* = 0·0750), with a minimum follow-up of 22.8 months. The protocol-determined boundary for significance (*p* = 0.0419) for the primary endpoint of OS was not reached. Of note, both the proportion of patients with grade 3/4 treatment-related AE and any grade of treatment-related AE leading to discontinuation were reduced in the nivolumab versus sorafenib groups (22% vs. 56%). In this trial, nivolumab had a manageable safety profile with no new safety signals observed.

The efficacy of combination therapies including nivolumab was been evaluated in the CheckMate 040 trial. Nivolumab plus ipilimumab was administered in patients from the previously-mentioned Checkmate 040 trial who had aHCC and were sorafenib refractory or sorafenib intolerant [28]. In this RCT, 148 patients were randomized into three dosing arms. The first group was administered nivolumab 1 mg/kg plus ipilimumab 3 mg/kg every three weeks for a period of four doses and the second group was treated with nivolumab 3 mg/kg plus ipilimumab 1 mg/kg every three weeks for a period of four doses, while both groups were later administered nivolumab 240 mg intravenously every two weeks. A third group was treated with nivolumab 3 mg/kg every two weeks plus ipilimumab 1 mg/kg every six weeks. With a median follow-up period of 30.7 months (IQR 29.9–34.7), the trial demonstrated a clear-cut clinical benefit among patients treated with nivolumab combined with ipilimumab, with a high ORR of 32% in group A, 31% in group B, and 31% in group C. In addition, patients in the first group achieved the highest complete response rate and had a median OS of 22.8 months. Twelve-month, 24-month, and 30-month OS was 61%, 48%, and 44%, respectively. A more recent trial evaluated the safety and efficacy of nivolumab combined with cabozantinib and ipilimumab in patients with aHCC [29]. A total of 71 sorafenib-naive or sorafenib–treated patients were randomized to treatment with either nivolumab plus cabozantinib (arm A) or nivolumab plus cabozantinib and ipilimumab (arm B) while treatment continued until intolerable toxicity or disease progression was demonstrated. The observed disease control rate was 81% for arm A and 83% for arm B patients, while the median progression-free survival was 5.5 months for arm A and 6.8 months for arm B patients. Median OS was not reached in either arm. Grade 3/4 treatment-related AE were 42% in the arm A and 71% in arm B patients, leading to discontinuation in one (3%) and seven (20%) patients, respectively.

### 2.3. Pembrolizumab

Pembrolizumab is an anti-PD-1 IgG4 mAb granted approval by the FDA in 2018 as a second-line treatment option for patients with aHCC previously treated with sorafenib and who present with progressive disease or high toxicity. The KEYNOTE-224 was the first phase 2 trial, recruiting 104 patients previously treated with sorafenib who were either intolerant to this treatment or demonstrated radiographic progression of their disease after treatment [30]. Patients were administered 200 mg pembrolizumab intravenously every three weeks for approximately two years or until demonstration of disease progression, unacceptable toxicity, patient withdrawal, or investigating team decision. An ORR of 17% was reported, while the median OS and progression-free survival (PFS) rates were 12.9 and 4.9 months, respectively [30].

Combination therapy including lenvatinib (a selective, multi-targeted TKI of VEGFR 1–3) and pembrolizumab has previously been successfully used in patients with advanced endometrial cancer [31]. Lenvatinib has the ability to inhibit the proneoangiogenic and immunosuppressive effects of tumor microenvironments, and such inhibition may maximize the clinical benefit of PD-1 antibodies by enhancing the antitumor immune response [32]. Using this rationale, a multicentric open-label study recruited 104 patients with aHCC who received lenvatinib and pembrolizumab. The reported median OS and PFS was 22 and 9.3 months, respectively, while grade ≥3 treatment-related AE occurred in 67% of patients.

Following the success of KEYNOTE-224 in demonstrating the antitumor activity and safety of pembrolizumab among patients with aHCC, a subsequent randomized, double-blind, placebo-controlled phase III trial sought to confirm the efficacy and safety of pembrolizumab plus best supportive care (BSC) versus placebo with BSC in patients with aHCC [33]. The KEYNOTE-240 trial randomized a total of 413 patients to pembrolizumab (*n* = 278) or placebo (*n* = 135). The ORR was 18.3% (95% CI, 14.0–23.4%) for pembrolizumab and 4.4% (95% CI, 1.6–9.4%) for placebo (*p* = 0.00007), while median OS was 13.9 months (95% CI, 11.6–16.0 months) in the pembrolizumab group and 10.6 months (95% CI, 8.3–13.5 months) in the placebo group (HR, 0.781; 95% CI, 0.611 to 0.998; *p* = 0.0238). Although the primary endpoints were not reached to achieve the predetermined statistical significance per specified criteria, the results of the trial were consistent with those reported in the KEYNOTE-224 trial, thus justifying a favorable risk-to-benefit ratio for pembrolizumab in patients with aHCC [33].

### 2.4. Tremelimumab

Tremelimumab is a fully human IgG2 mAb that binds to CTLA-4 and results in inhibition of B7-CTLA-4-mediated downregulation of T cell activation [34]. Tremelimumab has previously been demonstrated to induce a significant tumor response in a subgroup of patients with metastatic colorectal cancer and melanoma [35,36].

A phase II single arm, open-label, multicenter clinical trial aimed to test the antitumor and antiviral effect of tremelimumab in patients with HCC and HCV infection [37]. Twenty-one patients with aHCC confirmed by biopsy or non-invasive criteria and chronic HCV infection, Child-Pugh class A or B, and disease not amenable to percutaneous ablation or transarterial therapy were included in the trial. Treatment was largely well-tolerated, with few patients experienced disabling AE; no patient received systemic steroids and there were no treatment-related deaths [37]. In an intention-to-treat analysis including all 21 patients, the median time to progression was 6.48 months (95% CI 3.95–9.14) and the median OS was 8.2 months (95% CI 4.64–21.34). The 6- and 12-month survival were 64% and 43%, respectively. 

A phase I/II open-label randomized study of durvalumab combined with tremelimumab in aHCC patients noted promising outcomes. The phase I component of the trial included forty patients, 30% of whom had received no prior systemic therapy. Grade ≥ 3 related AE occurred in 20%, while only three patients discontinued treatment due to AEs. No unexpected safety signals with durvalumab and tremelimumab were seen in the study population [38]. In the completed version of the trial, a total of 332 patients with aHCC and progressive disease or intolerance during sorafenib were randomized to receive tremelimumab (300 mg) plus durvalumab (*n* = 75), durvalumab monotherapy (104), tremelimumab monotherapy (*n* = 69), or tremelimumab (75 mg) plus durvalumab (*n* = 85) [38]. Grade ≥ 3 treatment-related AE occurred in 37.8%, 20.8%, 43.5%, and 24.4% of patients, respectively. Discontinuation because of treatment-related AEs was similar across all groups at 10.8%, 7.9%, 13.0%, and 6.1%, respectively. Median OS was highest with tremelimumab (300 mg) plus durvalumab at 18.73 months (95% CI 10.78–27.27), 15.11 months (95% CI 11.33–20.50) with tremelimumab, 13.57 months (95% CI 8.74–17.64) with durvalumab and 11.30 months (95% CI 8.38–14.95) with tremelimumab (75 mg) plus durvalumab [38].

## 3. Predictors of Response

Administration of ICI has been associated with manifestation of AE leading to discontinuation, and in some cases disease hyperprogression [39,40]. To this end, establishment of biomarkers to highlight groups of patients prone to benefit from ICI treatment versus those who are at high risk for AE is of paramount importance. To date, there is no universally acknowledged biomarker to accurately predict response to ICI in patients with aHCC. The most common biomarkers evaluated by trials include PD-L1 expression, tumor mutational burden (TMB), microsatellite instability (MSI), and DNA damage repair (DDR) gene alterations [39].

In 2012, a phase 1 study aimed to assess the safety, anti-tumor activity, and pharmacokinetics of a specific fully human IgG4-blocking mAb directed against PD-1 in patients with advanced solid tumors including melanoma, non–small-cell lung cancer, castration-resistant prostate cancer, and renal-cell and colorectal cancer [41]. Based on data in the study, 17 patients with PD-L1 negative tumors did not respond to anti-PD-1 therapy, whereas 36% of patients with PD-L1--positive tumors had an objective response. PD-L1 expression was subsequently investigated as a potential biomarker to predict the efficacy of anti-PD-1/PD-L1 therapy [42]. Additional clinical studies demonstrated a strong association between high PD-L1 expression prior to anti-PD-1/PD-L1 therapy with improved ORRs and survival in patients with various cancers including non-small-cell lung cancer, neck squamous cell carcinoma, and melanoma [43,44,45]. In the case of aHCC, a French retrospective study aimed to characterize PD-L1 expression in a series of 217 surgically resected HCCs from 199 unselected patients with various underlying risks factors [46]. PD-L1 expression in either neoplastic or intratumoral inflammatory cells was correlated with high tumor aggressiveness [46]. Furthermore, researchers from the Checkmate040 trial sought to investigate several biomarkers within the inflamed tumor microenvironment with the aim of highlighting possible associations with enhanced efficacy of nivolumab in patients with aHCC [47]. Fresh and archival tumor samples from both the dose-escalation and dose-expansion phases were analyzed by immunohistochemistry and RNA sequencing to evaluate several inflammatory gene expression signatures. Interestingly, the analysis demonstrated a survival benefit in patients with increased PD-L1 expression. More specifically, the median OS for patients with PD-L1 ≥ 1% was 28.1 months (95% CI 18.2–N/A), while it was 16.6 months (95% CI 14.2–20.2) for patients with PD-L1 <1% (*p* = 0.032) [47]. While PD-L1 expression may be a reliable predictor of tumor response, a number of limitations, including the unstandardized cut-off value used to define positivity, the temporal and spatial heterogeneity of PD-L1 expression, and the complexity of the methods needed for analysis, hinder its wider use [42].

Tumor mutational burden (TMB) is defined as a measure of the total number of mutations per coding area of a tumor genome [39,42]. Higher TMB tumors are thought to express more neoantigens, hence allowing for an enhanced immune anti-tumor response and subsequently an improved response to immunotherapy. In patients treated with immunotherapy for advanced melanoma, non-small-cell lung carcinoma, and several diverse tumors, TMB and high expression of neoantigens have been demonstrated to predict a higher ORR or prolonged survival [42,48]. HCC is characterized by an above-average TMB with repeated formation of neoantigens. As such, these patients can be expected to have a good response to PD-1/PD-L1 inhibition [49]. Nonetheless, as mutations may or may not be immunogenic, TMB is not a very reliable or widely used predictor of response. 

Microsatellite instability (MSI) is a phenotype of hyper-mutations caused by the loss of DNA mismatch–repair (MMR) activity. MSI was the first predictive biomarker for anti-PD-1 blockage approved by the FDA [42]. More specifically, the use of pembrolizumab was approved by the FDA in 2017 for the treatment of pediatric and adult patients with unresectable or metastatic MSI-high or deficient MMR (dMMR) solid tumors that had progressed following first-line standard of care treatment at the time [50]. In previously conducted trials, patients with -MSI-high tumors had upregulation of multiple immune checkpoints, including PD-1, thus making PD-1/PD-L1 blockade a logical targeting treatment approach [42,51]. Impressive results of pembrolizumab among patients with dMMR or MSI-high tumors after progression from prior chemotherapies have been demonstrated in several trials (KEYNOTE-016, 164, 012, 028, and 158), with a cumulative overall response rate of 39.6% (95% CI 31.7–47.9) [52] Nevertheless, MSI-high HCC tumors seem to be rare, ranging from 0.8 to 3% of patients [51]. A recent study from Japan aimed to investigate the incidence of MSI-high tumors in 82 consecutive patients with aHCC who had progressed after standard of care treatment [53]. MSI-high tumors were noted in only 2.4% of patients, with one patient showing complete response to pembrolizumab for over ten months and another patient not responding to treatment.

Activated β-catenin signaling has been suggested as a promising biomarker of resistance to immunotherapy and has been associated with immune exclusion in HCC [54]. De Galaretta et al. used a novel genetically engineered mouse HCC model to investigate how different genetic alterations affect immune surveillance as well as the response to immunotherapies [54]. β-catenin activation in HCC tumor cells was an important mechanism of immune escape that conferred resistance to anti-PD-1 therapies; these data warrant validation in prospective studies.

## 4. Immune Related Adverse Events

In spite of the significant clinical benefits of immunotherapy among patients with aHCC, administration of ICI can lead to a wide spectrum of immune-related AE [55,56]. While treatment with ICI is generally well tolerated, the potential for life-threatening severe AE and/or irreversible organ damage remains [57].The most frequently encountered AE following ICI administration affect the skin, gastrointestinal tract, lungs, endocrine system (including thyroid), adrenal and pituitary glands, kidneys, and the nervous, hematologic, and cardiovascular systems. The most common immune-related AE in aHCC patients who receive anti-CTLA-4 treatment are skin rash, fatigue, and diarrhea, as well as disorders of thyroid hormone secretion and liver function derangement [55,56,58]. Patients treated with anti-PD-1 antibodies experience fewer immune-related AE than anti-CTLA-4 therapies. Notably, due to the complementary mechanisms of anti-CTLA-4 and anti-PD-L1 therapies, patients receiving combined therapy are at higher risk of experiencing cumulative AE and toxicity compared to individuals receiving monotherapy with ICI [59]. 

Deciding which patients should interrupt therapy as a result of an AE is challenging. Additionally, as the optimal duration of ICI therapy is not always clearly defined, the decision to resume therapy after resolution of toxicity is complicated. Early trials used a benchmark of one year; later trials increased this to two years of therapy or continued treatment with ICI until disease progression was demonstrated or patients became intolerant [60]. Among patients with mild AE, administration of ICI may continue with the precondition of close monitoring, whereas special attention must be paid to patients with moderate/severe AE. Grade 3/4 AE may be associated with critical organ dysfunction and decline in quality of life, and several deaths attributed to severe AE have been reported [61]. To this end, prompt detection and appropriate management of these toxicities is of critical importance [56]. Zhang et al. performed a meta-analysis of twelve RCTs comprising 5775 patients treated with ipilimumab for solid tumors, with the aim of determining the overall risk of fatal AEs [61]. The authors reported that the pooled incidence of fatal AEs was 1.1% (95% CI, 0.6–1.9%); treatment with ipilimumab was associated with a statistically significant increased risk of fatal AEs with a pooled Peto OR of 2.3 (95% CI, 1.4–3.6; *p* < 0.001).

A growing body of evidence suggests that there is a potential association between immune-related AEs and improved long-term outcomes, as these two events are speculated to share similar immunological bases [62,63]. Theoretically, patients experiencing higher grade AEs are expected to have elevated T-cell activity, and subsequently to enjoy improved antitumor outcomes compared with individuals who experience a lower-grade AE. This effect has been widely reported in patients treated with ICI for a plethora of cancers including gastrointestinal tumors, melanoma, and head and neck tumors [62,63,64,65]. A recently published retrospective single-center study from Singapore aimed to investigate the association between immune-related AEs and the efficacy of ICI treatment in patients with aHCC [58]. Monotherapy and combination therapy were administered to 82.7% and 17.3% of patients, respectively. The most frequently encountered all-grade AEs were dermatological (47%), hepatobiliary (14.3%), and endocrine (9.5%). Similar to previous studies, patients treated with combination therapy were more prone to experience a grade ≥3 AE than those who received monotherapy (31% vs. 10.8%, *p* = 0.009). Moreover, patients with any grade AE had a longer median OS versus patients without AE (16.2 months, 95% CI 13.9–20.7 vs. 4.6 months, 95% CI 3.2–5.7; HR 0.45, 95% CI 0.31–0.66; *p* < 0.001). Thus, the authors concluded that the presence of AEs might serve as a promising prognostic, as patients with more severe AEs and multiorgan involvement are shown to have a better prognosis. The key to achieving the optimal long-term outcomes for these patients is, however, early treatment of AEs with the use of systemic corticosteroids. Close monitoring throughout treatment is therefore critical to detect as well as to promptly and adequately treat AEs. 

## 5. Future Perspectives

### 5.1. Evaluation of ICI Efficacy Based on Underlying Liver Disease

The beneficial effect of ICI administration in patients with aHCC has been validated by several clinical trials. Nevertheless, recently published data have suggested that survival may not be improved in patients with non-viral HCC [66,67]. For example, a recently published study reported a marked accumulation of CD8^+^PD1^+^ T cells in NASH-HCC mice [67]. Although this effect was initially expected to predict a good response to anti-PD-1 treatment, NASH-HCC mice experienced higher hepatic tissue damage and HCC progression. In contrast, non-NASH-HCC mice had tumor regression following anti-PD-1 treatment. 

A subsequent meta-analysis of three key RCTs demonstrated that despite immunotherapy leading to an increased OS in the overall population (HR 0.77; 95 % CI 0.63–0.94) and among patients with HBV- or HCV-related HCC (HR 0.64; 95% CI 0.48–0.94), there was not a similar effect among patients with non-viral HCC (HR 0.92; 95% CI 0.77–1.11) [66]. Although the authors acknowledged that these outcomes were derived from trials with variable lines of treatment in patients with heterogeneous underlying liver diseases, the results suggest that the administration of ICI in patients with aHCC should be tailored based on underlying liver disease. For example, among 130 patients treated with anti-PD L1 immunotherapy for aHCC, patients with underlying NAFLD had a shorter OS versus patients with other HCC-related etiologies (5.4 vs 11 months, *p* = 0.023) [66].

### 5.2. Ongoing Trials

A number of ongoing clinical trials seek to assess the efficacy of ICI either as monotherapy or combination therapy for patients with aHCC, as summarized in Table 2. 

### 5.3. ICI in Patients with Resectable Disease

Perioperative administration of ICI in patients with early/resectable HCC disease is poorly reported, with only a very limited number of studies published to date. Studies evaluating the safety and efficacy of ICI in other types of cancers including non-small-cell lung cancer, dMMR colon, early-stage melanoma, and bladder cancers have, however, shown promising outcomes [68,69,70]. Notably, these studies have reported that patients with major pathologic response following ICI administration enjoyed better long-term outcomes postoperatively.

A single-arm phase 1b study evaluated the feasibility of neoadjuvant cabozantinib and nivolumab in patients with HCC, including patients outside of traditional resection criteria [71]. Of note, upfront surgical resection was not recommended for any of the included patients based on the initial multidisciplinary evaluation secondary to the presence of high-risk tumor features that historically predict poor outcomes with upfront surgical resection. Fifteen patients were included, 80% of whom subsequently underwent successful margin negative resection, while 42% patients had major pathologic responses. Despite the brief eight-week neoadjuvant treatment course, a significant proportion of patients demonstrated radiographic tumor changes in turn resulting in enhanced resectability, while no significant radiographic tumor progression was noted in any of the included patients over the treatment course [71]. Currently, two recruiting RCTs are assessing outcomes of neoadjuvant administration of ICI in patients with early and intermediate stage HCC. NCT04174781 is a phase II single-arm open-label study recruiting patients with BCLC A/B HCC with the aim of evaluating the safety and efficacy of sintilimab injection combined with TACE-DEB. The primary endpoint is PFS and the secondary endpoints include ORR, OS, duration of response, major pathological response rate, and R0 resection rate, as well as occurrence of AEs. NCT03510871 is another single-arm open-label trial recruiting patients with early and intermediate HCC and potentially eligible for curative surgery, and is aimed at evaluating the efficacy (i.e., tumor shrinkage), ORR, and down-staging of nivolumab plus ipilimumab as neoadjuvant therapy. Furthermore, tumor tissue and peripheral blood samples will be collected from recruited patients to evaluate for biomarkers for nivolumab plus ipilimumab immunotherapy. The primary endpoint of the trial is the percentage of subjects with tumor shrinkage after study drug treatment. NCT04658147 is an open-label phase 1 trial aiming to assess the feasibility and efficacy of perioperative nivolumab with or without relatlimab for patients with potentially resectable HCC. The primary endpoint is the proportion of patients who complete pre-operative treatment and proceed to surgery; secondary endpoints include the number of patients experiencing study drug-related toxicities, R0 resection, and pathologic response rates, as well as OS and DFS at 12, 18, 36, and 60 months.

A more recently published single center RCT aimed to evaluate whether nivolumab alone or combined with ipilimumab can be safely administered and induce increased immunological and clinical responses among patients with resectable HCC [72]. Twenty-seven patients were randomly assigned to nivolumab (*n* = 13) or nivolumab plus ipilimumab (*n* = 14). As expected, the nivolumab plus ipilimumab group experienced more grades 3/4 AE compared to the monotherapy group (43% vs. 23%); however, no patients in either group had any delay in surgical resection due to grade 3 or worse AE. The estimated median PFS was 9.4 months (95% CI 1.47–not estimable) in the nivolumab group and 19.53 months (2.33–not estimable) in the nivolumab plus ipilimumab group (HR 0.99, 95% CI 0.31–2.54) [72].

## 6. Conclusions

Notwithstanding the promising outcomes demonstrated by various trials in favor of immunotherapy, the survival of patients with aHCC remains dismal. Prompt identification of AE through close monitoring throughout the treatment period is critical for administering the appropriate treatment and minimizing the associated mortality. Additionally, reliable biomarkers that can distinguish which patients will benefit most from treatment immunotherapy are necessary to maximize the clinical benefit of ICI. Stratification of patients based on underlying liver disease may play a critical role in future studies and potentially lead to maximizing the beneficial effect of ICI. Further studies among patients with resectable HCC are warranted in order to elucidate whether immunotherapy can be transformed from palliative treatment in the unresectable advanced setting to a meaningful curative treatment option for patients with early disease.

## Figures and Tables

**Table 1 cancers-14-02018-t001:** Clinical trials assessing the safety and efficacy of ICI in patients with aHCC.

ICI	Type of Treatment	Trial Name	Intervention (*n* of Patients)	ORR (%) ^a^	OS ^b^ (Months)	Grade 3/4 AE (%)	Discontinuation Due to AE (%)
Atezolizumab	Combined therapy	GO30140	atezolizumab + bevacizumab (104)	36	17.1	53	17
		GO30140	atezolizumab + bevacizumab (60) vs. atezolizumab (59)	20 vs. 17	NR vs. NR	68 vs. 41	3 vs. 2
	Comparative RCT	IMbrave150	atezolizumab + bevacizumab (336) vs. sorafenib (165)	27.3 vs. 11.9	67.2% vs. 54.6% ^f^	56.5 vs. 55.1	15.5 vs. 10.3
Nivolumab	Monotherapy	CheckMate 040	nivolumab (48 ^c^, 214 ^d^)	15, 20	15, NR	25, NR	4, 8
	Combined therapy	CheckMate 040	nivolumab + ipilimumab (50, 49, 49) ^e^	32, 31, 31	22.8, 12.5, 12.7	10, 4, 2	18, 6, 2
		CheckMate 040	nivolumab + cabozatinib (35) vs. nivolumab + ipilimumab + cabozatinib (36)	17 vs. 26	NR vs. NR	42 vs. 71	3 vs. 20
	Comparative RCT	CheckMate 459	nivolumab (371) vs. sorafenib (372)	15 vs. 7	16.4 vs. 14.7	22 vs. 49	4 vs. 8
Pembrolizumab	Monotherapy	KEYNOTE-224	pembrolizumab (104)	17	12.9	25	4.8
	Combined therapy	NCT03006926	lenvatinib + pembrolizumab (104)	46	22	67	14
	Comparative RCT	KEYNOTE-240	pembrolizumab (278) vs. placebo (135)	18.3 vs 4.4	13.9 vs. 10.6	52.7 vs. 46.3	17.2 vs. 8.1
Tremelimumab	Monotherapy	NCT01008358	tremelimumab (21)	17.6% ^g^	8.2	NR	NR
	Combined therapy	NCT02519348	Durvalumab + tremelimumab (75) vs. durvalumab (104) vs. tremelimumab (69) vs. durvalumab + tremelimumab (84)	24.0% vs. 10.6% vs. 7.2% vs. 9.5%	18.7 vs. 15.1 vs. 13.6 vs. 11.3	43 vs. 56 vs. 46 vs. 50	10.8% vs. 7.9% vs. 13.0% vs. 6.1%

ICI: Immune checkpoint inhibitor; ORR: objective response rate; OS: overall survival; AE: adverse events; RCT, randomized controlled trial; NR: not reported; ^a^: as per mRECIST criteria; ^b^: median, ^c^: dose escalation group, ^d^: dose expansion group, ^e^: nivolumab 1 mg/kg plus ipilimumab 3 mg/kg,, followed by nivolumab (group a); nivolumab 3 mg/kg plus ipilimumab 1 mg/kg, followed by nivolumab (group b); or nivolumab 3 mg/kg plus ipilimumab (group c); ^f^: 12-month; ^g^: partial response.

**Table 2 cancers-14-02018-t002:** Ongoing clinical trials assessing outcomes of patients with advanced HCC (aHCC) treated with immune checkpoint inhibitor mono- or combined therapy.

Phase	Type of Treatment	Trial Identifier	Intervention	Study Population	Primary Endpoint	Estimated Completion Date
1/2	Monotherapy	NCT03630640	nivolumab	Patients with aHCC naïve to systemic therapy	AE, sAE, ORR	2024
2	Monotherapy	NCT02702414	pembrolizumab	Patients with aHCC	ORR	2022
1/2	Monotherapy	NCT02940496	pembrolizumab	Patients with aHCC (2nd line)	DLT	2022
3	Monotherapy	NCT03412773	tislelizumab	Patients with aHCC (1st line)	OS	2022
3	Combined	NCT03755791	atezolizumab + cabozatinib	Patients with aHCC (1st line)	OS, PFS	2023
1/2	Combined	NCT03170960	atezolizumab + cabozatinib	Patients with aHCC (1st line)	ORR, MTD	2022
3	Combined	NCT03298451	durvalumab + tremelimumab	Patients with aHCC (1st line)	OS	2024
3	Combined	NCT03764293	camelizumab + apatinib	Patients with aHCC (1st line)	OS, PFS	2022
2	Combined	NCT03033446	nivolumab +TARE	Patients with aHCC	RR	2022
1/2	Combined	NCT04170556	nivolumab + regorafenib	Patients with aHCC progressing after 1st line therapy	AE	2023
2	Combined	NCT04310709	nivolumab + regorafenib	Patients with aHCC naïve to systemic therapy	RR	2023
3	Combined	NCT04039607	nivolumab + ipilimumab	Patients with aHCC (1st line)	OS	2025
2	Combined	NCT03781960	nivolumab + abemaciclib	Patients with aHCC	ORR	2022
2	Combined	NCT03841201	nivolumab + lenvatinib	Patients with aHCC	ORR, AEs	2022
2	Combined	NCT04050462	nivolumab + cabrializumab	Patients with aHCC	ORR	2024
3	Combined	NCT03713593	pembrolizumab + lenvatinib	Patients with aHCC (1st line)	OS, PFS	2023
3	Combined	NCT04246177	pembrolizumab + lenvatinib + TACE	Patients with aHCC (1st line)	OS, PFS	2029
2	Combined	NCT03519997	pembrolizumab + bavituximab	Patients with aHCC	ORR	2023
2	Combined	NCT03316872	pembrolizumab + SBRT	Patients with aHCC	ORR	2023
2	Combined	NCT04696055	pembrolizumab + regorafenib	Patients with aHCC previously treated with PD-1/PD-L1 ICI	ORR	2024

AE: adverse events; sAE: serious adverse events; ORR: objective response rate; DLT: dose limiting toxicity; TARE: transarterial radio embolization; RR: response rates; OS: overall survival; PFS: progression-free survival; SBRT: stereotaxic body radiation therapy; MTD: maximum tolerated dose.

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
