# Peer review of "Current Landscape of Immune Checkpoint Inhibitor Therapy for Hepatocellular Carcinoma"

_cancers, 2022, doi:10.3390/cancers14082018_

Round 1

Reviewer 1 Report

Manuscript ID: cancers-1677073 Title: Current landscape of immune checkpoint inhibitor therapy for hepatocellular carcinoma

 The authors reviewed many previous papers concerning immune checkpoint therapy for HCC. I think that this review should be well written, organized, and therefore easily understood by most readers. If the authors could consider several points listed below and modify /add some descriptions, this review would be more informative and useful for the readers.

Major point
 I think it is necessary to mention the relationship between the underlying liver disease and the effects of ICI. For example, the following papers provide important insights into the effects of ICI in each etiology.

Nature. 2021 Apr;592(7854):450-456.  doi: 10.1038/s41586-021-03362-0.  Epub 2021 Mar 24.

Nature. 2021 Apr;592(7854):444-449.  doi: 10.1038/s41586-021-03233-8.  Epub 2021 Mar 24.

Minor point 1. The keywords should be: hepatocellular carcinoma; liver cirrhosis; immune checkpoint inhibitors 2. In page 2 line 75-77. The citation is inappropriate because it contains gastric cancer literature rather than lung cancer literature. 3. In page 4 line 148. 585 may be 58.5%. 4. In page 5 line 194-195. “sorafeniband’ must be changed to “sorafenib and”. 5. In page 6 line 226. “igG2” must be changed to “IgG2”. 6. In page 6 line 247. (n=104) 7. In page 8 line 74. “moths” must be changed to “months”.
8. Table2. Change 20224 to 2022  

Author Response

Remark 1. The authors reviewed many previous papers concerning immune checkpoint therapy for HCC. I think that this review should be well written, organized, and therefore easily understood by most readers. If the authors could consider several points listed below and modify /add some descriptions, this review would be more informative and useful for the readers.

We would like to thank the reviewer for the positive comments.

Remark 2. Major point
 I think it is necessary to mention the relationship between the underlying liver disease and the effects of ICI. For example, the following papers provide important insights into the effects of ICI in each etiology. Nature. 2021 Apr;592(7854):450-456.  doi: 10.1038/s41586-021-03362-0.  Epub 2021 Mar 24. Nature. 2021 Apr;592(7854):444-449.  doi: 10.1038/s41586-021-03233-8.  Epub 2021 Mar 24.

We would like to thank the reviewer. As requested, we have now added a separate section in our future perspectives in which we review the suggested articles. We have also added a relevant comment in the conclusion section to highlight the important role of underlying liver disease to stratify patients who receive ICI.

Remark 3. Minor point 1. The keywords should be: hepatocellular carcinoma; liver cirrhosis; immune checkpoint inhibitors 2. In page 2 line 75-77. The citation is inappropriate because it contains gastric cancer literature rather than lung cancer literature. 3. In page 4 line 148. 585 may be 58.5%. 4. In page 5 line 194-195. “sorafeniband’ must be changed to “sorafenib and”. 5. In page 6 line 226. “igG2” must be changed to “IgG2”. 6. In page 6 line 247. (n=104) 7. In page 8 line 74. “moths” must be changed to “months”. 8. Table2. Change 20224 to 2022  

We have revised the manuscript and proposed references accordingly.

Reviewer 2 Report

Dr Nikolaos Machairas and co-authors present a very comprehensive review on the current landscape of immune checkpoint inhibitor therapy for hepatocellular carcinoma. Their review is based on recent clinical trials assessing patients with advanced HCC and highlights predictors of response and immune-related adverse events. Additionally, they summarise outcomes of perioperative administration of ICI in patients with resectable HCC.

Overall, their paper describes a novel topic that surely deserves attention. The review is very well written, extremely comprehensive and is based on Level 1 evidence studies (RCT). There are just a couple of minor issues I’d like to bring to the authors’ and editor’s attention:

  • What I somehow miss is a paragraph based on current recommendations illustrating current indications of ICI for both resectable and unresectable HCC. I was wondering if it could be possible to implement this aspect in some way.
  • Similarly, a comparison between “standard” treatments for HCC (TKIs) vs ICI for unresectable HCC.

Author Response

Remark 1. Dr Nikolaos Machairas and co-authors present a very comprehensive review on the current landscape of immune checkpoint inhibitor therapy for hepatocellular carcinoma. Their review is based on recent clinical trials assessing patients with advanced HCC and highlights predictors of response and immune-related adverse events. Additionally, they summarise outcomes of perioperative administration of ICI in patients with resectable HCC. Overall, their paper describes a novel topic that surely deserves attention. The review is very well written, extremely comprehensive and is based on Level 1 evidence studies (RCT).

We would like to thank the reviewer.

Remark 2. What I somehow miss is a paragraph based on current recommendations illustrating current indications of ICI for both resectable and unresectable HCC. I was wondering if it could be possible to implement this aspect in some way.

As requested, we have added a paragraph on the current EASL recommendations related to ICI utilization. Regarding immunotherapy use among patients with resectable disease, data are still limited and thus there are currently no formal recommendations. In turn, administration of ICI should be limited to clinical trials until more solid evidence of its beneficial effect is published.

Remark 3. Similarly, a comparison between “standard” treatments for HCC (TKIs) vs ICI for unresectable HCC.

We respectfully believe that all the major trials comparing ICI monotherapy (Checkmate 459 & Keynote 240) or combination therapy (Imbrave 150 & Checkmate 040) vs. Sorafenib have been provided in detail in the text, as well as in Table 1.